# Synthesis and Self-Assembly of Multistimulus-Responsive Azobenzene-Containing Diblock Copolymer through RAFT Polymerization

**DOI:** 10.3390/polym11122028

**Published:** 2019-12-06

**Authors:** Po-Chih Yang, Yueh-Han Chien, Shih-Hsuan Tseng, Chia-Chung Lin, Kai-Yu Huang

**Affiliations:** Department of Chemical Engineering and Materials Science, Yuan Ze University, Chung-Li, Taoyuan City 32003, Taiwan; s1055224@mail.yzu.edu.tw (Y.-H.C.); s1055221@mail.yzu.edu.tw (S.-H.T.); s1065204@mail.yzu.edu.tw (C.-C.L.); s1065222@mail.yzu.edu.tw (K.-Y.H.)

**Keywords:** azobenzene, reversible addition-fragmentation transfer (RAFT), photoisomerization, *N*-isopropylacrylamide, sensing

## Abstract

This paper gathered studies on multistimulus-responsive sensing and self-assembly behavior of a novel amphiphilic diblock copolymer through a two-step reverse addition-fragmentation transfer (RAFT) polymerization technique. *N*-Isopropylacrylamide (NIPAM) macromolecular chain transfer agent and diblock copolymer (poly(NIPAM-*b*-Azo)) were discovered to have moderate thermal decomposition temperatures of 351.8 and 370.8 °C, respectively, indicating that their thermal stability was enhanced because of the azobenzene segments incorporated into the block copolymer. The diblock copolymer was determined to exhibit a lower critical solution temperature of 34.4 °C. Poly(NIPAM-*b*-Azo) demonstrated a higher photoisomerization rate constant (*k_t_* = 0.1295 s^−1^) than the Azo monomer did (*k_t_* = 0.088 s^−1^). When ultraviolet (UV) irradiation was applied, the intensity of fluorescence gradually increased, suggesting that UV irradiation enhanced the fluorescence of self-assembled cis-isomers of azobenzene. Morphological aggregates before and after UV irradiation are shown in scanning electron microscopy (SEM) and dynamic light scattering (DLS) analyses of the diblock copolymer. We employed photoluminescence titrations to reveal that the diblock copolymer was highly sensitive toward Ru^3+^ and Ba^2+^, as was indicated by the crown ether acting as a recognition moiety between azobenzene units. Micellar aggregates were formed in the polymer aqueous solution through dissolution; their mean diameters were approximately 205.8 and 364.6 nm at temperatures of 25.0 and 40.0 °C, respectively. Our findings contribute to research on photoresponsive and chemosensory polymer material developments.

## 1. Introduction

The potential applications for stimulus-responsive polymers have expanded remarkably in recent years and now include fluorescent chemosensors, biological probes, and drug delivery vehicles that utilize pH, temperature, light, humidity, electric field, or ionic strength [1,2,3,4,5,6,7]. Due to their photoinduced isomerization, optical data storage, holographic grating properties, liquid crystal anisotropy, and photomechanical bending, azobenzenes—which can be employed as versatile photoresponsive materials—have been the focus of considerable research [8,9,10,11]. Azobenzene-containing polymers can be used in the fields of mechanics, self-organized structuring, mass transport, optics, photonics, materials science, and bionics [12,13]. Photodriven changes in molecular alignment can play an important role in the future. Applications as an artificial muscle and an optical plastic motor are in progress [14]. However, the fluorescent properties of azobenzene chromophores have been investigated in few studies because the chromophores exhibit non-fluorescent or low fluorescence quantum yield (*Φ*_PL_) under highly efficient E-Z (trans-cis) isomerization in the photoexcited state. Fluorescence is detectable when molecular interaction or self-organization suppresses azobenzene photoisomerization [15,16]. Kawashima et al. [17] demonstrated that the azobenzenes exhibit fluorescent emission when the intramolecular N–B interaction (N and B mean Nitrogen and Boron atom, respectively) is incorporated through the employment of the bis(pentafluorophenyl)boryl group. An azobenzene chromophore was reported by Bandara et al. [18] that had strong intramolecular hydrogen bonds and intense fluorescence. Using the reverse addition-fragmentation transfer (RAFT) technique, Zhu et al. synthesized terminal-functionalized polystyrene that exhibited an azobenzene structure; in addition, enhanced fluorescence emission was achieved following ultraviolet (UV) irradiation [19]. In our previous study [20], we examined how, upon UV irradiation, a terminal electron-donating methoxy (–OCH_3_) group of azobenzene chromophores had significantly more intense fluorescence than an electron-withdrawing nitro (–NO_2_) group did.

Amphiphilic block copolymers can form cylinders, micelles, vesicles, and other aggregates when the solvent interacts selectively with hydrophobic and hydrophilic blocks [21,22]. Previously executed studies have employed RAFT and atom transfer radical polymerization techniques to derive azobenzene-containing amphiphilic block copolymers with both the self-assembling characteristics that block copolymers exhibit in solutions and the photoresponsive properties that azobenzene polymers exhibit; the aggregates that were obtained further affected their fluorescence properties [23,24,25,26]. The first study to discover that azobenzene-containing block copolymer micelles in mixture solution are fluorescent was Zhao et al. [27]. Liu et al. [28] prepared a fluorescent polymer (PCN250) that is responsive to temperature and pH and exhibits higher fluorescence intensity at higher temperature when the pH is 4–10. Jellyfish-like “breathing” vesicles containing azobenzene have been reported by Dong et al. [24,25] to exhibit pH- and visible-light-dependent on–off switchable fluorescence. Recently, we employed microwave heating and RAFT polymerization to obtain an amphiphilic azobenzene-based photoresponsive diblock copolymer (poly(4-acetoxystyrene)-*block*-poly[6-(4-methoxy-azobenzene-4’-oxy) hexyl acrylate] (poly(StO_54_-*b*-Cazo_9_)) that demonstrates rapid microphase separation, high photoresponsive response, and an enantiotropic smectic A mesophase in copolymer [29].

Well known for its response to temperature, poly(*N*-isopropylacrylamide) (poly(NIPAM)) switches between states of being water insoluble and water soluble in response to its degree of order at various temperatures. Numerous scholars have investigated this material because it has a noteworthy lower critical solution temperature (LCST) of approximately 32.0 °C in water [30,31,32,33]. Scholars have also recently devoted substantial attention to poly(NIPAM)-containing polymers because they can be widely applied in drug delivery, sensors, and temperature-targeted therapy systems [34,35,36]. The polymers‘ LCST can be modulated through their copolymerization with hydrophilic and hydrophobic comonomers or the incorporation of salts and surfactants [37,38,39,40]. Few studies appear to have investigated adjustment of the fluorescence of fluorophores in poly(NIPAM)-containing polymers by using optically and thermally induced aggregates [41,42]. Tang et al. [43] demonstrated that above the LCST, the interchain or intrachain hydrogen bonds within oligo(ethylene glycol) methacrylate-containing poly(NIPAM) are favored to form emissive polymer aggregates. Watanabe et al. [44] synthesized several random copolymers comprising 4-phenylazophenyl methacrylate and NIPAM (poly(4-phenylazophenyl methacrylate)-*ran*-poly(*N*-Isopropylacrylamide) P(AzoMA-*r*-NIPAm)) and reported that at a bistable temperature, the copolymers underwent a photoinduced phase transition that was reversible. Oriol et al. [45] reported a new set of photo-responsive supramolecular amphiphilic block copolymers that were synthesized on the basis of the association between 4-isobutyloxyazobenzene and 2,6-diacylaminopyridine units through multiple H-bonding interactions. Wang et al. [46] prepared a tetraphenylethene-containing poly(NIPAM) exhibiting aggregation-induced emission when assembled into nanoparticles in water, and the fluorescence intensity was lower when the temperature was higher.

Two-step RAFT polymerization techniques were employed in this study to obtain a NIPAM-containing dual-stimulus-responsive (i.e., temperature and light) diblock copolymer (poly(NIPAM-*b*-Azo)) with azobenzene units as comonomers. We investigated the structural effect of the polymeric structure’s azobenzene group on the poly(NIPAM-*b*-Azo) solution’s self-assembly and thermal, optical, and chemosensory properties under various conditions and temperatures. The kinetics study of photoisomerization revealed larger rate constants for the diblock copolymer than for the azobenzene monomer (Azo). We noted that the fluorescence of the Z-isomers self-assembled from azobenzenes in poly(NIPAM-*b*-Azo) solution was enhanced upon UV irradiation. NIPAM macromolecular chain transfer agent (macro-CTA) and poly(NIPAM-*b*-Azo) were found to have LCST values of approximately 32.6 and 34.4 °C, respectively. We discovered that poly(NIPAM-*b*-Azo) responded highly selectively, rapid, and sensitively to Ru^3+^ and Ba^2+^ in a tetrahydrofuran (THF)–H_2_O mixture. Moreover, the addition of the hydrophobic azobenzene unit to poly(NIPAM) induced a microphase separation, which suggests that this polymer may have wide application to rapid microphase separation and high-potential optical response.

## 2. Experimental

### 2.1. Materials

Scheme 1 and Scheme 2 illustrate the synthetic routes for the azobenzene-based monomer and its corresponding diblock copolymer, respectively. By executing a previously reported process [20,47], we synthesized 4-hydroxy-4′-methoxy-azobenzene. We purified NIPAM through recrystallization from *n*-hexane before use and purified as well as distilled THF from sodium prior. Sodium or calcium hydride was used to dry *N*,*N*-dimethylformamide (DMF), which then underwent distillation under reduced pressure and storage over a 4-Å molecular sieve before use. We purchased from chemical companies 2-[2-(2-chloroethoxy)ethoxy]ethanol (Tokyo Chemical Industry, Tokyo, Japan, 96.0%), acryloyl chloride (Aldrich, Saint Louis, MO, USA, 97.0%), potassium carbonate (Showa, Tokyo, Japan, 99.5%), triethylamine (Aldrich, Saint Louis, MO, USA, 99.5%), *N*-isopropylacrylamide (Tokyo Chemical Industry, Tokyo, Japan, 98.0%), 2,2′-azoisobutyronitrile (AIBN; Aldrich, Saint Louis, MO, USA, 98.0%), 2-(dodecylthiocarbonothioylthio)-2-methylpropionic acid (Aldrich, Saint Louis, MO, USA, 98.0%), chloride (Ag^+^, Ba^2+^, Mg^2+^, Ni^2+^, Ca^2+^, Mn^2+^, Fe^3+^, Al^3+^, and Ru^3+^) or nitrate salts (K^+^ and Li^+^) of metal ions (K^+^, Li^+^, Ag^+^, Ba^2+^, Mg^2+^, Ni^2+^, Ca^2+^, Mn^2+^, Fe^3+^, Al^3+^, and Ru^3+^), and other reagents; we used them as received.

### 2.2. Measurements

A Bruker AMX-500 spectrometer (Billerica, MA, USA) was employed to obtain nuclear magnetic resonance (NMR) spectra; CDCl_3_ was employed as the solvent, with tetramethylsilane being employed as the internal standard; we reported chemical shifts (*δ*) in units of ppm. Moreover, we applied a Heraeus CHN–O rapid elemental analyzer (Darmstadt, Germany) for elemental analysis. We measured the polydispersity index (PDI) and weight-average molecular weight (*M*_w_) of polymers through gel permeation chromatography (GPC; model CR4A from Shimadzu, Kyoto, Japan). THF was the GPC eluent, polystyrene standards (1000–136,000 g/mol) were employed to calibrate the instrument, and the rate of elution was 1.0 mL/min. We performed thermal analysis using the Perkin Elmer DSC 7 differential scanning calorimeter (Waltham, MA, USA) under N_2_ atmosphere and with a 20 K/min scanning rate. Again under N_2_ atmosphere, we executed thermogravimetric analysis (TGA) using a Perkin Elmer TGA-7 thermal analyzer (Waltham, MA, USA) and 20 K/min heating rate. A Jasco V-670 spectrophotometer (Jasco corporation, Tokyo, Japan) was employed to obtain UV-visible (UV-vis) absorption spectra; the apparatus was equipped with a Peltier thermostatted single cell holder (air cooled) for temperature control measurements. The LCST of the polymers, for a polymer concentration of 2.5 mg/mL in water, was detected by the change in transmittance at 600 nm. The temperature at which the transmittance-temperature curve passed through 50% transmittance was determined to be the LCST of the polymer aqueous solution. We determined the fluorescence properties of the polymers by using an OBB Quattro II fluorescence spectrophotometer (manufacturer, city, country). With poly(9,9-dihexylfluorene) as the standard (*Φ*_PL_ = 1.0) and at room temperature, the polymers solution’s fluorescence quantum yield (*Φ*_PL_) were derived. The Magic Droplet Model 100SB instrument (Sindatek Instrument Co., Ltd., Taipei, Taiwan) was employed to perform dynamic light scattering (DLS); the instrument had a thermostated sample chamber, with the scattering being executed using a 4 mW He-Ne laser (*λ* = 632.8 nm). Scanning electron microscopy (SEM) images were recorded using a JEOL JSM 5600 SEM system (JEOL, Osaka, Japan). The polymer concentration was 0.5 mg mL^−1^ for SEM observation.

### 2.3. Synthesis of Intermediate and Monomer (Scheme 1)

#### 2.3.1. Synthesis of 2-[2-(2-(4-Methoxy-Azobenzene-4’-oxy)Ethoxy)Ethoxy]Ethanol (1)

We heated a mixture of the following at 100 °C for 30 min; 4-hydroxy-4′-methoxy-azobenzene (0.45 g, 2.0 mmol), potassium iodide (0.25 mg, 0.0015 mmol), potassium carbonate (0.55 g, 4.0 mmol), and DMF (25 mL). Subsequently, 2-[2-(2-chloroethoxy)ethoxy]ethanol (0.4 g, 2.4 mmol) was then added dropwise. The derived mixture was then stirred for 24 h. Immediately after being cooled to room temperature, we poured the mixture into ice water under stirring, after which we extracted it with dichloromethane. The organic layer was subjected to washing twice with water, followed by drying over anhydrous magnesium sulfate and concentration under reduced pressure. Recrystallization was employed to purify the crude product from ethanol and thereby obtain 1, a white solid (82.2%). ^1^H NMR (CDCl_3_, 500 MHz) was used to analyze this solid, with the following values for the chemical shift *δ*_H_ (ppm): 3.60–3.62 (t, 2H, –CH_2_–), 3.68–3.74 (m, 6H, –CH_2_–), 3.86 (s, 3H, –OCH_3_–), 3.87–3.89 (t, 2H, –CH_2_–), 4.18–4.23 (t, 2H, –CH_2_–), 6.96–7.02 (d, 4H, aromatic, Ar–H), and 7.83–7.89 (d, 4H, aromatic, Ar–H)

#### 2.3.2. Synthesis of 2-[2-(2-(4-Methoxy-Azobenzene-4’-Oxy)Ethoxy)Ethoxy]Ethyl Acrylate (2; Azo)

We dissolved a solution containing 1 (0.36 g, 1.0 mmol) and a catalytic amount of 2,6-di-*tert*-butyl-*p*-cresol in triethylamine (0.18 g, 2.5 mmol) and dichloromethane (10 mL). An ice bath was employed to cool the solution, to which was then added (dropwise) acryloyl chloride (0.15 g, 1.5 mmol) dissolved in dichloromethane (5 mL); subsequently, the mixture was stirred vigorously for 2 h under N_2_ atmosphere. When the reaction was complete, we poured the solution into ice water and recrystallized twice from ethanol to afford 2 (68.7%). ^1^H NMR (acetone-*d*_6_, 500 MHz): *δ*_H_ (ppm) = 3.64–3.67 (t, 4H, –CH_2_–), 3.68–3.72 (t, 2H, –CH_2_–), 3.86–3.89 (t, 2H, –CH_2_–), 3.90 (s, 3H, –OCH_3_), 4.23–4.29 (t, 4H, –CH_2_–), 5.88 (dd, 1H, CH_2_=CH), 6.16 (dd, 1H, CH_2_=CH), 6.36 (dd, 1H, CH_2_=CH), 7.07–7.14 (d, 4H, aromatic, Ar–H), and 7.84–7.90 (d, 4H, aromatic, Ar–H). Anal. Calcd. (%) for C_22_H_2__6_N_2_O_6_: C, 63.76; H, 6.32; and N, 6.76. Found: C 63.85; H, 6.28; and N, 6.81.

### 2.4. Synthesis of Polymers (Scheme 2)

#### 2.4.1. Synthesis of NIPAM-Functionalized Macro-CTA

NIPAM monomer (1.80 g, 16.0 mmol), 2-(dodecylthiocarbonothioylthio)-2-methylpropionic acid (the CTA; 36.4 mg, 0.1 mmol), AIBN (3.2 mg, 0.02 mmol), and THF (2.0 mL) were placed in a Schlenk tube and magnetically stirred. The molar ratio of monomer/CTA/AIBN was 800/5/1. Dry N_2_ was used to purge the mixture, after which any dissolved oxygen was removed by subjecting the mixture to three freeze-pump-thaw cycles. We sealed the tube under vacuum, and it was then immersed in an oil bath at 60 °C for 24 h. At the end of this period, we cooled the mixture to room temperature. The solvent was removed, and excess cold diethyl ether was employed to filter the residue and thus precipitate out the polymer. The precipitate was subjected to vacuum drying to obtain the product (44.8%). The glass transition and decomposition temperatures *T*_g_ and *T*_d_ were 186.6 and 351.8 °C, respectively, whereas the number-average molecular weight *M*_n_ = 6.98 × 10^3^ g/mol and the PDI = 1.25. ^1^H NMR (CDCl_3_, 500 MHz) was employed to analyze the product, with *δ*_H_ (ppm) = 0.85 (s, –CH_3_), 0.95–1.30 (br, –CH_2_–, –CH_3_), 1.34–2.45 (br, –CH_2_–CH–), 3.48 (br, –S–CH–), and 3.90–4.08 (br, –NHCH(CH_3_)_2_).

#### 2.4.2. Synthesis of Azobenzene-Functionalized Diblock Copolymer (poly(NIPAM-*b*-Azo))

We synthesized the diblock copolymer poly(NIPAM-*b*-Azo) using the second-step RAFT method with the macro-RAFT agent being NIPAM macro-CTA and AIBN being the initiator in THF at 60 °C. A typical procedure was as follows. Into a Schlenk tube were placed Azo (132.0 mg, 0.32 mmol), NIPAM macro-CTA (0.84 g), AIBN (3.2 mg, 0.02 mmol), and 2 mL THF. Degassing was achieved using three freeze-pump-thaw cycles, after which the tube was vacuum sealed and subjected to oil bath immersion for 24 h at 60 °C. THF was employed to dilute the mixture, which was subsequently dropped into diethyl ether. Product purification was achieved through two rounds of reprecipitation from THF to cold diethyl ether. Subsequently, overnight, the product was subjected to vacuum oven drying at room temperature. We determined the block length ratio of the copolymer from the NMR spectrum. The yield of poly(NIPAM-*b*-Azo) was 52.0%; *T*_g_ = 159.8 °C; *T*_d_ = 370.8 ^o^C, *M*_n_ = 9.96 × 10^3^ g/mol; and PDI = 1.24. ^1^H NMR (acetone-*d*_6_, 500 MHz) was employed with δ_H_ (ppm) = 0.86 (s, –CH_3_), 1.04–2.42 (br, –CH_2_–, –CH_3_, –CH_2_–CH–), 3.55–3.80 (br, –CH_2_–, –CH_3_), 3.88–4.12 (br, –NHCH(CH_3_)_2_), and 6.81–7.35 (br, aromatic, Ar–H).

### 2.5. Fluorescent Titration with Metal Ions

We employed THF solutions to performed fluorescent titration experiments. The stock solutions (1.0 × 10^−3^ M) comprised deionized water containing chloride (Ag^+^, Ba^2+^, Mg^2+^, Ni^2+^, Ca^2+^, Mn^2+^, Fe^3+^, Al^3+^, and Ru^3^^+^) or nitrate salts (K^+^ and Li^+^). A stock solution was added to a polymer-solution-containing test tube to conduct individual titrations processes. Immediately after preparation and thorough mixture of the test solution, optical measurements were obtained. The final polymer concentration was 1.0 × 10^−6^ M. In addition, the estimated water: THF ratio 1:99.

## 3. Results and Discussion

### 3.1. Synthesis of Azobenzene Monomer

The route through which the azobenzene-functionalized monomer (labeled 2 (Azo) in the Appendix A) was synthesized is illustrated in Scheme 1. We obtained the azobenzene-functionalized precursor 1,2-[2-(2-(4-methoxy-azobenzene-4’-oxy)ethoxy)ethoxy]ethanol, at 100 °C through nucleophilic substitution reaction of 2-[2-(2-chloroethoxy)ethoxy]ethanol with 4-hydroxy-4′-methoxy-azobenzene in the presence of potassium carbonate. We employed recrystallization from ethanol to purify the crude product and thereby obtain 1; the yield of this intermediate was 82.2%. We subsequently prepared 2-[2-(2-(4-methoxy-azobenzene-4’-oxy)ethoxy)ethoxy]ethyl acrylate (2; Azo) in a 68.7% yield by using an esterification reaction at 0 °C between intermediate 1 and acryloyl chloride in the presence of triethylamine. Through the execution of ^1^H NMR, we confirmed the chemical structure and constitutional composition of the synthesized compounds. The ^1^H NMR spectra derived for azobenzene compounds 1 and 2 (Azo) as well as the corresponding structural assignments are presented in Appendix A. The characteristic chemical shifts *δ* = 7.83–7.89 (*H*_a_) and 6.96–7.02 ppm (*H*_b_) were attributed to aromatic doublet proton signals from the azobenzene intermediate 1; conversely, the shift *δ* = 4.18–4.23, 3.87–3.89, 3.68–3.74, 3.60–3.62, and 3.86 ppm were ascribed to the triplet proton signals of aliphatic methylene (*H*_c_, *H*_d_, *H*_f +_
_g_, and *H*_h_) and methoxy group singlet protons (*H*_e_), respectively.

The presence of the vinyl group doublet of doublets (dd; *H*_c_, *H*_d_, and *H*_e_) in protons at 5.88–6.36 ppm (Appendix A) confirmed that the azobenzene-functionalized monomer (Azo) was synthesized successfully. Signals from other aromatic protons were discovered at 7.07–7.90 ppm. Additionally, peaks were discovered at 4.23–4.29, 3.86–3.89, 3.68–3.72, and 3.64–3.67 ppm in the ^1^H NMR spectrum that was derived for Azo; these peaks were assigned to the *H*_f_, *H*_h_, *H*_i_, and *H*_j_ methylene protons, respectively, of the azobenzene unit’s ethylene glycol groups. The methoxy group singlet protons (*H*_g_) exhibited peaks at 3.90 ppm. Elemental analysis was also employed to confirm the synthesized monomer’s chemical structure.

### 3.2. Polymer Synthesis and Thermal Properties

RAFT polymerization was employed to synthesize NIPAM macro-CTA, which was reacted in THF at 60 °C by employing NIPAM monomer in a 800:5:1 [M]_0_:[CTA]_0_:[AIBN]_0_ molar ratio. The RAFT polymerization findings are presented in Table 1, and Scheme 2 displays the routes to synthesizing NIPAM macro-CTA and poly(NIPAM-*b*-Azo). We used ^1^H NMR spectroscopy and GPC to determine the structure of the NIPAM macro-CTA. The PDI and *M*_n_ of the NIPAM macro-CTA were 1.25 and 6.98 × 10^3^ g mol^−1^, respectively (Table 1).

The NIPAM macro-CTA ^1^H NMR spectrum, recorded in CDCl_3_, is displayed in Appendix A. We calculated the degree of polymerization (DP) of the NIPAM macro-CTA on the basis of the integral ratio of methine protons H_a_ (at 3.90–4.08 ppm), derived from the NIPAM monomer, to methyl protons H_d_ in the CTA agent (at 0.85 ppm). A DP of 59.5 was obtained, and this revealed that the RAFT agent moiety was attached to the ends of the NIPAM macro-CTA. The *M*_n(NMR)_ of the polymer was evaluated through the data in Appendix A and the following Equation to be 7087 g mol^−1^:*M*_n(NMR)_ = (3(*H*_a_/*H*_d_) × *M*_n,__NIPAM_) + *M*_n,CT__A_,(1)
where *M*_n,__NIPAM_ and *M*_n,CT__A_ are the molecular weight of the NIPAM monomer and CTA, respectively, and *H*_a_ and *H*_d_ are the integrals of methine protons (–CH) and methyl protons, respectively. Using GPC, we determined *M*_n_ to be 6980 g mol^−1^. Thus, the percentage of NIPAM macro-CTA chains that had their ends capped with CTA groups was discovered to be equal to roughly 98.5%.

The ^1^H NMR spectrum of the poly(NIPAM-*b*-Azo) in acetone-*d*_6_ is displayed in Appendix A. The integrated peak areas of the methine protons (*H*_c_: 3.88–4.12 ppm) were compared with those of aromatic protons (*H*_a_ and *H*_b_: 6.81–7.35 ppm) in Azo to calculate the molar ratio of NIPAM to Azo units as being 59.5:7.4. Finally, the diblock copolymer was approximately formulated as poly(NIPAM_60_-*b*-Azo_7_).

The decomposition temperature (*T*_d_) values of the NIPAM macro-CTA and poly(NIPAM-*b*-Azo) were 351.8 and 370.8 °C, respectively; this revealed that the thermal stability was enhanced because azobenzene segments were incorporated into the block copolymer.

### 3.3. Optical Properties of Azobenzene Monomer and Polymer

UV absorption spectroscopy was performed to examine the kinetic rate of photoisomerization of Azo and poly(NIPAM-*b*-Azo) in THF solution (1.0 × 10^−6^ M); this revealed how the photoreactivity of the synthesized diblock copolymer was structurally affected by the Azo monomer. Appendix A presents the UV-vis spectra of the Azo monomer for several durations of 365 nm light irradiation; the figure also indicates the stability of these spectra in darkness. Strong UV-vis absorption bands were observed in the spectra at approximately 357 nm, which were attributable to the π–π* transition in chromophores’ E-isomer. Moreover, weak absorption bands at approximately 450 nm were engendered by an n–π* transition in the E-isomer. The differences between the absorption spectra reveal E-Z isomerization in the Azo groups. The monomer achieved an E-Z photostationary state within 35 s of the beginning of irradiation. Under UV irradiation, the E-Z isomerization rate and time of the photostationary states were affected by the potential energy profiles between the E-Z isomers. In darkness, it took 180 min for the monomer to become sufficiently thermally stable for Z-E isomerization. However, poly(NIPAM-*b*-Azo) entered a steady state for Z-E isomerization after 300 min (Figure 1), showing slow thermal isomerization of the diblock copolymer from the Z to E form [48,49]. These results also suggest that the Z-E isomerization of poly(NIPAM-*b*-Azo) was restrained in segmental mobility by surrounding polymer chains.

We further investigated the kinetics of the photoisomerization process. Figure 2 presents the data obtained and the curves fit to these data. The following function was employed in the fitting of spectral variation:*Ln*((*A*_eq_ − *A*_t_)/(*A*_eq_ − *A*_0_)) = –*kt*,(2)
where *A*_eq_ is the absorbance in the UV-vis spectra for a photostationary state at 357 nm, *A*_0_ and *A*_t_ are the 357 nm absorbance prior to and after UV irradiation, respectively, and *k* is the photoisomerization rate constant. The first-order kinetic rate was obtained from the E-Z photoisomerization kinetic curves, which are displayed in Figure 2. For Azo and poly(NIPAM-*b*-Azo), the derived *k* values were 8.8 × 10^−2^ and 1.295 × 10^−^^1^ s^−1^, respectively. These results suggest that the diblock copolymer’s large rate constants were a consequence of the low Azo block ratio of the polymer chain and the limited free volume available for E-Z photoisomerization of Azo chromophores within the diblock copolymer.

### 3.4. Thermoresponsive Properties of Polymers

Figure 3 illustrates the LCST behavior of the NIPAM macro-CTA and poly(NIPAM-*b*-Azo). The corresponding transmittance of the polymers at 25 and 40 °C is displayed in the inset of Figure 3. In aqueous solution, poly(NIPAM) generally collapsed, with dense globule chains forming at approximately 32 °C. The LCST of the NIPAM macro-CTA was 32.6 °C, consistent with the findings reported in the literature for the LCST of poly(NIPAM) [30,31,32,33]. The transparent aqueous solution containing the NIPAM macro-CTA became opaque at temperatures higher than the LCST, as illustrated in Figure 3 displaying the results obtained from UV spectroscopic analyses. However, the LCST was 34.4 °C, almost 2 °C higher than that of the NIPAM macro-CTA (32.6 °C), implying that the higher LCST of the poly(NIPAM-*b*-Azo) micelle compared with that of the NIPAM macro-CTA homopolymer was potentially caused by steric repulsion of the poly(NIPAM) chains tethered to the micelle‘s hydrophobic azobenzene core [50].

Using DLS, we determined the hydrodynamic diameters (*D*_h_) of the NIPAM macro-CTA and poly(NIPAM-*b*-Azo) nanoassemblies at two temperatures: one lower than and one higher than the LCST (i.e., 25 and 40 °C; Appendix A and Figure 4). The *D*_h_ of the NIPAM macro-CTA at a temperature higher than the LCST increased from 25–70 nm to 150–950 nm, as expected and indicating that micelles had aggregated because the poly(NIPAM) block became dehydrated. We observed irregular morphological micelles of poly(NIPAM) at 40 °C in aqueous solution. Nevertheless, we noted that the *D*_h_ values of the poly(NIPAM-*b*-Azo) micelles were approximately 160–270 and 220–560 nm at 25 and 40 °C, respectively. Therefore, adding the hydrophobic azobenzene unit to the diblock copolymer induced a microphase separation, resulting in regular micelles and smaller morphologies (Figure 4).

Subsequently, the influence of temperature on the fluorescence of the poly(NIPAM-*b*-Azo) in solution at 25 and 40 °C was investigated. Figure 5 shows a dramatically higher fluorescence intensity of poly(NIPAM-*b*-Azo) at temperatures higher than the LCST. By employing poly(9,9-dihexylfluorene) as a reference (*Φ*_PL_ = 1.0), we estimated the polymer’s fluorescence quantum yield (*Φ*_PL_). The *Φ*_PL_ values of poly(NIPAM-*b*-Azo) in aqueous solution were 8.7 × 10^−2^ and 2.7 × 10^−1^ at 25 and 40 °C, respectively; this increase in *Φ*_PL_ was because the poly(NIPAM) block underwent a thermally induced coil-to-globule transition, preventing rotation of the azobenzene chromophore and enhancing the fluorescence intensity [51].

### 3.5. Photoresponsive Properties of Diblock Copolymer

Changes often occur in the molecular dipole moment of azobenzene when it undergoes E-Z photoisomerization, and these changes influence azobenzene’s aggregation behavior. This can achieve UV-controllable fluorescence of poly(NIPAM-*b*-Azo). Figure 6 illustrates the dependence on 365 nm irradiation duration of the fluorescence spectra of poly(NIPAM-*b*-Azo) in THF solution. When the poly(NIPAM-*b*-Azo) solution was excited at 350 nm, we observed weak fluorescence at approximately 402–426 nm. Irradiation with UV light caused increased fluorescence intensity within 1 min of commencement of irradiation, and the fluorescence intensity gradually increased after 30 min of continuous irradiation. Notably, the *Φ*_PL_ of the poly(NIPAM-*b*-Azo) solution increased from 2.9 × 10^−2^ upon UV irradiation for 0 min to 4.0 × 10^−2^ for 1 min and to 5.2 × 10^−2^ for 30 min. These findings revealed that UV irradiation results in enhancement of the fluorescence of self-assembled Z-isomers of azobenzenes [52,53] and also that azobenzene moiety packing was more dense when they were in their Z form than their E form, causing enhancement of fluorescence intensity following UV irradiation.

We further prepared the morphological aggregates by adding water to dilute the homogeneous solution of the diblock copolymer. Poly(NIPAM-*b*-Azo) formed micellar aggregates in response to UV irradiation, as confirmed by SEM observations and dynamic light scattering (DLS) analyses. Figure 7 illustrates the SEM images of the micellar aggregates obtained using the poly(NIPAM-*b*-Azo) (0.5 mg mL^−1^) with a THF/H_2_O volume ratio of 8/2. The *D*_h_ of the spherical micelles was approximately 420.5 nm, as shown in Figure 7a, indicating that the spherical micelles had formed because of the relatively short hydrophobic azo block (at approximately 10.5%) in the diblock copolymer. The spherical micelles further worsened when exposed to continuous UV irradiation for 10 min, and the micelles gradually formed irregular aggregates. The concentration of these spherical micelles decreased when exposed to continuous UV irradiation for 10 min, and the micelles gradually formed irregular aggregates. After irradiating the aggregates under 30 min of UV continuous irradiation, we observed fewer micelles in the solution (Figure 7b), indicating that the polymeric aggregates underwent photoisomerization from E to Z-isomers. The results suggest that the construction variation of the morphological aggregates can be attributed to the increased hydrophilicity of the azo block, thereby causing the azo chromophores to shrink. Consequently, the micelle aggregates were collapsed and broken when irradiated. It also means that the observed fluorescence enhancement upon UV irradiation (i.e., Figure 6) can be related to changes in sizes and types of morphological aggregates.

Figure 8a presents the DLS distribution of spherical micelles formed from the poly(NIPAM-*b*-Azo), with the *D*_h_ being approximately 257.7 nm. After UV irradiation for 10 min, there was an obvious increase in the average diameter to 405.4 nm. Moreover, the *D*_h_ of the poly(NIPAM-*b*-Azo) after UV irradiation increased from 170–420 to 250–650 nm, indicating that the hydrophobic E-isomer of the azobenzene transformed into the strong polar Z-isomer under UV irradiation, and the tight cores of regular E-isomer with an ordered array transformed to the Z-isomer, which packed into the loose cores as the unordered array [54].

### 3.6. Ion Sensing Properties

Oxygen-containing ligands (e.g., ethylene glycol) are generally used for combining alkali or alkaline metal cations to generate complexes [55]. Poly(NIPAM-*b*-Azo) was complexed with numerous metal ions (Li^+^, Mg^2+^, Al^3+^, K^+^, Ca^2+^, Mn^2+^, Fe^3+^, Ni^2+^, Ag^+^, Ba^2+^, and Ru^3^^+^), and the effect of the specific ion on its fluorescence spectra in THF‒H_2_O solution was determined. When we separately titrated the polymer solution with Ag^+^, Al^3+^, Fe^3+^, Mg^2+^, Mn^2+^, or Ni^2+^, the photoluminescence (PL) intensity decreased slightly, which are displayed in Figure 9a. However, the PL intensity of the polymer was significantly enhanced when Ba^2^^+^ or Ru^3+^ ions were used, indicating that the oxygen atoms of ethylene glycol in the azobenzene unit and amide C=O groups of NIPAM blocked complexation with these two ions [56]. The polymer’s Photoluminescence (PL) response profiles (i.e., *I*/*I*_o_) when it was in the presence of metal ions (concentration of 5.0 × 10^−5^ M) are displayed in Figure 9b. The *Φ*_PL_ of the poly(NIPAM-*b*-Azo) solution with a concentration of 1.0 × 10^−6^ M increased from 2.4 × 10^−2^ to 7.5 × 10^−2^ for Ba^2^^+^ ions and to 5.4 × 10^−2^ for Ru^3^^+^ ions (concentration of 5.0 × 10^−^^5^ M); thus the polymer has high selectivity toward these two cations. Our results suggest that the polymer’s PL intensity enhancement can be attributed to the coordination of the ethylene glycol ligands binding to Ba^2^^+^ and Ru^3+^ ions and to the suitable match of ion radius to cavity size, leading to the generation of a macrocycle, bearing adjacent crown ether rings and azobenzene groups, and closed azobenzene packing. Pang et al. [57] employed density functional theory and reactive molecular dynamics to demonstrate that macrocycles with azobis-(benzo-18-crown-6) and alkaline metal ions faced each other when Ba^2^^+^ ions were complexed. They also indicated that the amide functionality that was stabilized within the cavity was provided by not only chelation of its C=O group to a metal complexed to one of the macrocycle’s oligo(ethylene glycol) loops but also hydrogen bonding between the oxygen atoms of the other loops and its NH protons [58]. These results suggest that ligands’ diameter, metal ion charge density, the ion–dipole interaction, and chelating capability strongly affect fluorescence, leading to the chelating ability variation observed in this study. Consequently, the charge (Ba^2^^+^: 0.89; Ru^3+^: 2.2) and diameter (Ba^2^^+^: 2.70 Å; Ru^3+^: 1.64 Å) of Ba^2^^+^ and Ru^3+^ ions are inherent characteristics affecting the strong coordination of these ions that accompany crown ether ring units. Therefore, poly(NIPAM-*b*-Azo) can be used as a polymer chemosensor for the sensing of cations.

Appendix A illustrates the *D*_h_ of the poly(NIPAM-*b*-Azo) aggregates in solution at 25 °C. When the polymer solution was chelated separately with Ba^2+^ or Ru^3+^ ions, the *D*_h_ of the polymer increased from 205.8 nm (Figure 4a) to 259.6 nm for Ba^2+^ ion and to 225.0 nm for Ru^3+^ ion at an ion concentration of 1.0 × 10^−4^ M. We observed irregular morphological aggregates in the range of approximately 70–430 and 100–370 nm for Ba^2+^ and Ru^3+^ ions, respectively. The results suggest that the polymer’s irregular aggregates were attributed to the coordination of the ethylene glycol ligands binding to Ba^2+^ and Ru^3+^ ions. The irregular morphological sizes may also have originated from the formation of other complexes in solutions and the repulsive forces between the polymer chains.

## 4. Conclusions

In this study, we synthesized a new azobenzene-functionalized amphiphilic diblock copolymer (poly(NIPAM_60_-*b*-Azo_7_)) through RAFT polymerization and evaluated its optical and sensory characteristics. The *M*_n_ values of NIPAM macro-CTA and poly(NIPAM_60_-*b*-Azo_7_), obtained using GPC, were 6.98 × 10^3^ and 9.96 × 10^3^ g mol^−1^, respectively; their PDIs were 1.25 and 1.24, respectively. Poly(NIPAM_60_-*b*-Azo_7_) had a higher photoisomerization rate constant (1.295 × 10^−1^ s^−1^) than the Azo monomer did (8.8 × 10^−2^ s^−1^). The NIPAM macro-CTA and diblock copolymer had LCSTs of approximately 32.6 and 34.4 °C, respectively. DLS revealed that incorporation of hydrophobic azobenzene in the diblock copolymer could induce microphase separation and the formation of regular micelles. UV irradiation led to changes in DLS distributions and changes of the micellar aggregates of poly(NIPAM_60_-*b*-Azo_7_). PL titrations demonstrated that the copolymer exhibited high sensitivity and enhanced intensity toward Ru^3+^ and Ba^2+^ ions, which could be attributed to the coordination of ethylene glycol ligands binding to these two cations because the ion radius is commensurate with the cavity size.

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
