# Peer review of "Synthesis and Self-Assembly of Multistimulus-Responsive Azobenzene-Containing Diblock Copolymer through RAFT Polymerization"

_polymers, 2019, doi:10.3390/polym11122028_

Round 1

Reviewer 1 Report

Reviewer report on Manuscript Draft entitled ‘Synthesis and Self-Assembly of  Multistimulus-Responsive Azobenzene-Containing Diblock Copolymer through RAFT Polymerization’

In this research authors describe multistimulus-responsive sensing and self-assembly behavior of a novel amphiphilic diblock copolymer through a two-step reverse addition-fragmentation transfer (RAFT) polymerization technique. PL titrations demonstrated that the copolymer exhibited high sensitivity and enhanced intensity toward Ru3+ and Ba2+ ions, which could be attributed to the coordination of ethylene glycol ligands binding to these two cations because the ion radius is commensurate with the size of formed cavity.

This manuscript is relatively well written and investigations are well addressed and are very interesting, from scientific (polymer chemistry, optics,  microphase separation)  and technological points of view. The research is in scope of the journal. Therefore, the manuscript can be published after some corrections and improvements:

In Introduction, Discussion, Conclusions more advanced overview of Azobenzene-based copolymers in optical devices (e.g. Experimental and Theoretical Investigations of an Electrochromic Azobenzene and 3,4-Ethylenedioxythiophene-Based Electrochemically Formed Polymeric Semiconductor. ChemPhysChem 2018, 19, 2735– 2740.) is required.

In some future trends can be presented with possible applicability of here reported Azobenzene-based polymers for some practical purposes.

In figures 2-4 and S4, error bars, which are representing experimental errors re missing.

Author Response

Response to Reviewer 1 Comments

Comments and Suggestions for Authors

Reviewer report on Manuscript Draft entitled ‘Synthesis and Self-Assembly of Multistimulus-Responsive Azobenzene-Containing Diblock Copolymer through RAFT Polymerization’. In this research authors describe multistimulus-responsive sensing and self-assembly behavior of a novel amphiphilic diblock copolymer through a two-step reverse addition-fragmentation transfer (RAFT) polymerization technique. PL titrations demonstrated that the copolymer exhibited high sensitivity and enhanced intensity toward Ru3+ and Ba2+ ions, which could be attributed to the coordination of ethylene glycol ligands binding to these two cations because the ion radius is commensurate with the size of formed cavity.

This manuscript is relatively well written and investigations are well addressed and are very interesting, from scientific (polymer chemistry, optics, microphase separation) and technological points of view. The research is in scope of the journal. Therefore, the manuscript can be published after some corrections and improvements:

Response: The authors would like to thank the reviewer for his/her comment. Truly, his/her insights contribute to the improvement of the edited version of our manuscript. We have carefully read the reviewers’ comments and revised our manuscript according to their suggestions. The detailed responses to the comments from the reviewers are listed as follows, and the revised places are marked with red letters in the revised manuscript. We hope that you and the reviewers will be satisfied our revisions.

Point 1: In Introduction, Discussion, Conclusions more advanced overview of Azobenzene-based copolymers in optical devices (e.g. Experimental and Theoretical Investigations of an Electrochromic Azobenzene and 3,4-Ethylenedioxythiophene-Based Electrochemically Formed Polymeric Semiconductor. ChemPhysChem 2018, 19, 2735– 2740.) is required.

Response 1: The authors agree with the reviewer’s comment. It is well-known that organic electrochromic semiconducting polymers, which are capable of changing their color upon applied electrochemical potential, which already are being successfully applied in smart windows, switchable mirrors and displays. 3,4-Ethylenedioxythiophene (EDOT)-based conjugated polymers seems to be ideal candidates for electrochromic materials for their response to electric field, and PEDOT switches between various oxidation and reduction states. In the reference the reviewer mentioned (i.e., ChemPhysChem 2018, 19, 2735), the main-chain polymeric structures are EDOT, and EDOT and thiophene units are also good conjugated structure or π-linker in electrochemical materials. However, our main chain of our synthesized polymer is polyethylene (i.e., polyacrylate derivatives), not conjugated units. The electrochemical properties and conductivity of polyacrylates are very poor. Due to the living RAFT polymerization, we only obtained the limited molecular weight and low PDI value of diblock copolymer. Besides, the synthesized azobenzene only had the main absorption region at 330-420 nm. Although azobenzenes are versatile photoresponsive materials under ultraviolet irradiation, we believe that such azobenzene copolymer as main structure may not be used efficiently in electrochromic materials.

We also agree that theoretical calculations or experiments are very important in research. Density functional theory (DFT) calculations or other software are used to evaluate the frontier molecular orbitals and semiconducting properties of the electrochromic polymer. Our research laboratory does not have the DFT calculation software, because of the limited funding and research background. In this manuscript, we only investigated the structural effect of the polymeric structure’s azobenzene group on the diblock copolymer solution’s self-assembly and thermal, optical, and chemosensory properties under various conditions. We also studied the kinetics study of photoisomerization for the diblock copolymer, not chromic responsive properties of optical devices. We look forward to having the opportunity to cooperate with other research groups or set up any optical modules/devices in the future. If there is something unsuitable, please accept our apologies and understand our difficult situation.

References: 1. Ramanavicius et al.; ChemPhysChem 2018, 19, 2735–2740. 2. Chahma et al.; Synth. Met. 2011, 161, 1532–1536.

Point 2: In some future trends can be presented with possible applicability of here reported Azobenzene-based polymers for some practical purposes.

Response 2: Thanks for the reviewer’s positive consideration and helpful comments. We agreed the reviewer’s valuable suggestions. We have provided the following statements of possible applicability for azobenzene-based polymers in the section of Introduction. “Azobenzene-containing polymers can be used in the fields of mechanics, self-organized structuring, mass transport, optics, photonics, materials science, and bionics. Photodriven changes in molecular alignment can play an important role in the future. Applications as an artificial muscle and an optical plastic motor are in progress”.

References: 1. Emoto et al.; Polymers 2012, 4, 150–186. 2. Fedele et al.; Biomater. Sci. 2018, 6, 990–995. 3. Mauro et al.; J. Mater. Chem. B 2019, 7, 4234–4242.

Point 3: In Figures 2-4 and S4, error bars, which are representing experimental errors are missing.

Response 3: Thanks for the reviewer’s helpful comments. We have provided the corresponding error bars of Figures 2-4 and S4-S5 in the revised manuscript.

Thank you very much for your time and consideration. I believe these revisions have radically improved the essay’s argument and clarity-thanks to the editors’ and reviewers’ thoughtful recommendations. This manuscript has been edited by Wallace Academic Editing, and is considered to be improved in grammar, punctuation, spelling, verb usage, sentence structure, general readability, writing style, and native English usage to the best of the editor’s ability. Please let me know if there is anything further I can do; in particle, I can return deleted material to the text for clarity.

Reviewer 2 Report

Authors report on the synthesis and responsive properties of an azobenzene-containing diblock copolymer. The manuscript is in general well written and organised. The topic well fits to the scope of this journal. The Tables and figures support the explanations in the text. The work will likely be of interest in the field, but I suggest that the following points are addressed before publication:

1) since the authors emphasized self-assembly in the title, I expect to read more about how the diblock copolymer is engineered: the authors observed that UV irradiation enhanced the fluorescence of self-assembled cis-isomers; such change could be due to formation of different aggregates and the authors should check for changes in the size and morphology of the self-assembled aggregates upon 365 nm light irradiation (i.e. using a water/THF mixture solution) by TEM and DLS;

2) coordination of Ba2+ and Ru3+ ions, respectively, might also affect the morphology of aggregates and the authors should, as above, use the same experiments (TEM, DLS) to investigate that;

3) the manuscript requires a more up-to-date references, only 4 of 50 references are from the period of 2016-2019;

4) page 3, under the Materials section: the authors used chloride or nitrate salts of different metal ions for the ion sensing properties study, however the authors should be more precise and report exactly which chlorides and nitrates salts were used.

In summary, the manuscript requires a major revision to address all the above concerns prior to further consideration for publication in this journal. 

Author Response

Response to Reviewer 2 Comments

Comments and Suggestions for Authors

Authors report on the synthesis and responsive properties of an azobenzene-containing diblock copolymer. The manuscript is in general well written and organised. The topic well fits to the scope of this journal. The Tables and figures support the explanations in the text. The work will likely be of interest in the field, but I suggest that the following points are addressed before publication:

Response: The authors would like to thank the reviewer for his/her comment. Truly, his/her insights contribute to the improvement of the edited version of our manuscript. We have carefully read the reviewers’ comments and revised our manuscript according to their suggestions. The detailed responses to the comments from the reviewers are listed as follows, and the revised places are marked with red letters in the revised manuscript. We hope that you and the reviewers will be satisfied our revisions.

Point 1: Since the authors emphasized self-assembly in the title, I expect to read more about how the diblock copolymer is engineered: the authors observed that UV irradiation enhanced the fluorescence of self-assembled cis-isomers; such change could be due to formation of different aggregates and the authors should check for changes in the size and morphology of the self-assembled aggregates upon 365 nm light irradiation (i.e. using a water/THF mixture solution) by TEM and DLS.

Response 1: Thanks for the reviewer’s positive consideration and comments. In this study, we examined the kinetic rate of the photoisomerization of Azo monomer and diblock copolymer (poly(NIPAM-b-Azo)) using UV absorption spectroscopy in a dilute THF solution to discern the structural effect of the Azo monomer on the photoreactivity of the synthesized diblock copolymer. We found that self-assembled Z-isomers (cis-isomers) of azobenzenes exhibit fluorescence enhancement upon UV irradiation. Our kinetic rate of the photoisomerization was performed in THF solution. However, the transmission electron microscopy (TEM) samples are observed in a solid state, not in organic or aqueous solutions. Up to now, there are not any liquid cell TEM or TEM with liquid modes in Taiwan. Besides, the TEM with an accelerating voltage of 200 kV at approximately 200 oC will induce the formation of deformation or decomposition of the soft diblock copolymer. Furthermore, the lower critical solution temperature of polymer was about 34.4 oC. The heat of the TEM electron beam will cause the polymer chain to undergo conformational changes or cracking. However, in order to meet the expectations of the reviewer, we have tried to observe the photoisomerization behavior of the polymer in the solid state. We did our best to adjust and reduce the electron beam energy and recorded the morphological aggregate of the polymer. Although these images may not have high resolution, or even this photoisomerization or self-assembly phenomena in solid state are not consistent with those findings in solution, we hope that the reviewer can accept our changes. Herein, we only provided the convincing TEM data in the revised manuscript. The detailed statements of TEM images before and after UV irradiation were provided in the section 3.5 of the revised manuscript. (See lines 357–365)

Point 2: Coordination of Ba2+ and Ru3+ ions, respectively, might also affect the morphology of aggregates and the authors should, as above, use the same experiments (TEM, DLS) to investigate that.

Response 2: Thanks for the reviewer’s comments. As mentioned above, we used TEM to observe the morphological aggregates of diblock copolymer in solid states. However, we employed the photoluminescence titrations to investigate the ion sensing properties of polymer in THF‒H2O solution, not in solid states. So far, Taiwan does not have any liquid cell TEM or TEM with liquid mode. Therefore, we provided the particle size distributions of poly(NIPAM-b-Azo) polymers after chelation of Ba2+ and Ru3+ (Figure S5) in the revised manuscript. The metal ion concentration in water was 1.0 × 10-4 M. These findings revealed that the average hydrodynamic diameter of the diblock copolymer increased from 205.8 nm (Figure 4a) to 259.6 nm for Ba2+ ion (Figure S5a) and to 225.0 nm for Ru3+ ions (Figure S5b), indicating the polymer’s irregular aggregates can be attributed to the coordination of the ethylene glycol ligands binding to these two cations. These results also suggest that addition of the ions might induce the formation of other complexes in solutions or repulsive forces between the polymeric chains, resulting in irregular morphological sizes. Additionally, because the repeating units of polymer chain are different, each polymer segment may chelate with different amounts of metal ions, which was the reason that the particle size distributions were broad in DLS analyses. (See lines 409–416)

Point 3: The manuscript requires a more up-to-date references, only 4 of 50 references are from the period of 2016-2019.

Response 3: Thanks for the reviewer’s positive consideration and comments. We have added some up-to-date references and removed some old ones. In the revised manuscript, 21 of 57 references are from the period of 2016-2019. We expect reviewer to accept such changes we made.

 Point 4: Page 3, under the Materials section: the authors used chloride or nitrate salts of different metal ions for the ion sensing properties study, however the authors should be more precise and report exactly which chlorides and nitrates salts were used.

Response 4: Thanks for the reviewer’s helpful comments. Among all the salts we used in the photoluminescence titrations, except that K+ and Li+ are nitrates, the other ions are chlorides. We have corrected the original sentence to “The stock solutions (1.0 × 10-3 M) comprised deionized water containing chloride (Ag+, Ba2+, Mg2+, Ni2+, Ca2+, Mn2+, Fe3+, Al3+, and Ru3+) or nitrate salts (K+ and Li+)” in the Sections 2.1 and 2.5 of the revised manuscript.

In summary, the manuscript requires a major revision to address all the above concerns prior to further consideration for publication in this journal.

Response: Thank you very much for your time and consideration. I believe these revisions have radically improved the essay’s argument and clarity-thanks to the editors’ and reviewers’ thoughtful recommendations. This manuscript has been edited by Wallace Academic Editing, and is considered to be improved in grammar, punctuation, spelling, verb usage, sentence structure, general readability, writing style, and native English usage to the best of the editor’s ability. Please let me know if there is anything further I can do; in particle, I can return deleted material to the text for clarity.

Round 2

Reviewer 2 Report

The authors were asked to verify the morphology and sizes of the self-assembled aggregates by TEM and DLS before and after UV irradiation, experiments that are very common when dealing with photoisomerizable systems, as very often the trans-to-cis photoisomerization of azobenzene-containing polymers is followed by changes in sizes and morphologies of the aggregates, see for example Polymers 2016, 8, 183; RSC Adv. 2017, 38335 etc. The observed fluorescence enhancement upon UV irradiation described in the manuscript could be related to changes in sizes and types of aggregates and such option should be verified.

The authors provided TEM images of poly(NIPAM-b-Azo) before and after UV irradiation (Figuire 7) however they are of insufficient quality for a scientific purpose. Nevertheless, even such TEM images show change in morphologies of aggregates and I am surprised that the authors did not put more efforts to provide representative TEM images. Meanwhile, the fluorescence intensity enhancement of the system is, according to the authors, attributed to more dense packing of azobenzene moiety in Z form than their E form within the aggregates. Yet, the particle size distribution of the irradiated sample was not provided although the different size of aggregates before and after irradiation should support this theory.

Furthermore, the authors were asked to use the same experiments (TEM, DLS) to check the influence of coordination of Ba2+ and Ru3+ ions, respectively, on the morphology of aggregates. However, only the particle size distributions are showed (Fig. S5), while the morphology images are not provided.

In conclusion, I am not satisfied with the author’s reply as the needed additional experiments were not conducted completely.

Author Response

Response to Reviewer 2 Comments

Comments and Suggestions for Authors

Point 1: The authors were asked to verify the morphology and sizes of the self-assembled aggregates by TEM and DLS before and after UV irradiation, experiments that are very common when dealing with photoisomerizable systems, as very often the trans-to-cis photoisomerization of azobenzene-containing polymers is followed by changes in sizes and morphologies of the aggregates, see for example Polymers 2016, 8, 183; RSC Adv. 2017, 38335 etc. The observed fluorescence enhancement upon UV irradiation described in the manuscript could be related to changes in sizes and types of aggregates and such option should be verified.

Response 1: Thanks for the reviewer’s positive consideration and comments. We all agreed the reviewer’s opinion and suggestion. After reading the reviewer’s comments, we have provided the SEM/DLS data and detailed statements in the section 3.5 (i.e., lines 361-391) of the revised manuscript. In Figure 7a, we can observe the micellar aggregates using the poly(NIPAM-b-Azo) (0.5 mg mL-1) in the THF-H2O mixture. The volume of THF/H2O was 8/2. These findings revealed that the spherical micelles had formed because of the relatively short hydrophobic azo block (at approximately 10.5%) in the diblock copolymer. After UV irradiation, the concentration of these micelles decreased when exposed to continuous UV irradiation for 10 min, and the micelles gradually formed irregular aggregates, indicating that the polymeric aggregates underwent photoisomerization from E to Z-isomers. These results also suggest that the observed fluorescence enhancement upon UV irradiation (i.e., Figure 6 in the revised manuscript) can be related to changes in sizes and types of the morphological aggregates. In Figure 8, we also found that there was an obvious increase in the average diameter after UV irradiation, indicating that the hydrophobic E-isomer of the azobenzene transformed into the strong polar Z-isomer under UV irradiation, and the tight cores of regular E-isomer with an ordered array transformed to the Z-isomer, which packed into the loose cores as the unordered array.

We employed SEM instead of TEM to observe micellar aggregates. We can use SEM of sufficient resolution to observe these aggregates. We are very sorry about this point. We hope that the editor and the reviewer will be satisfied our revisions.

Point 2: The authors provided TEM images of poly(NIPAM-b-Azo) before and after UV irradiation (Figure 7) however they are of insufficient quality for a scientific purpose. Nevertheless, even such TEM images show change in morphologies of aggregates and I am surprised that the authors did not put more efforts to provide representative TEM images. Meanwhile, the fluorescence intensity enhancement of the system is, according to the authors, attributed to more dense packing of azobenzene moiety in Z form than their E form within the aggregates. Yet, the particle size distribution of the irradiated sample was not provided although the different size of aggregates before and after irradiation should support this theory.

Response 2:  See the Response 1 above.

 Point 3: Furthermore, the authors were asked to use the same experiments (TEM, DLS) to check the influence of coordination of Ba2+ and Ru3+ ions, respectively, on the morphology of aggregates. However, only the particle size distributions are showed (Fig. S5), while the morphology images are not provided.

Response 3: Thanks for the reviewer’s comments. We used TEM to observe the morphological aggregates of diblock copolymer in solid states. However, we employed the photoluminescence titrations to investigate the ion sensing properties of polymer in THF‒H2O solution, not in solid states. Therefore, the solid state results in TEM will be inconsistent and inconclusive with those liquid states in ion sensing. In the revised manuscript, we have provided the particle size distributions of poly(NIPAM-b-Azo) polymers after chelation of Ba2+ and Ru3+ (Figure S5) in the revised manuscript. The detailed statements are given in lines 426-423. These results suggest that addition of the ions might induce the formation of other complexes in solutions or repulsive forces between the polymeric chains, resulting in irregular morphological sizes.

In summary, the manuscript requires a major revision to address all the above concerns prior to further consideration for publication in this journal.

Response: Thank you very much for your time and consideration. We did the best to address all the above concerns of reviewer 2. We believe these revisions have radically improved the essay’s argument and clarity-thanks to the editors’ and reviewers’ thoughtful recommendations. Please let me know if there is anything further I can do; in particle, I can return deleted material to the text for clarity.
